# Abnormal Morphology and Synaptogenic Signaling in Astrocytes Following Prenatal Opioid Exposure

**DOI:** 10.3390/cells13100837

**Published:** 2024-05-14

**Authors:** Ethan B. Niebergall, Daron Weekley, Anna Mazur, Nathan A. Olszewski, Kayla M. DeSchepper, N. Radant, Aishwarya S. Vijay, W. Christopher Risher

**Affiliations:** Department of Biomedical Sciences, Joan C. Edwards School of Medicine, Marshall University, Huntington, WV 25701, USA; niebergall1@marshall.edu (E.B.N.); weekley49@marshall.edu (D.W.); mazura@marshall.edu (A.M.); olszewski2@marshall.edu (N.A.O.); deschepper@marshall.edu (K.M.D.); radant@marshall.edu (N.R.); vijay@marshall.edu (A.S.V.)

**Keywords:** astrocytes, synapses, opioids, prenatal opioid exposure, tripartite, buprenorphine

## Abstract

In recent decades, there has been a dramatic rise in the rates of children being born after in utero exposure to drugs of abuse, particularly opioids. Opioids have been shown to have detrimental effects on neurons and glia in the central nervous system (CNS), but the impact of prenatal opioid exposure (POE) on still-developing synaptic circuitry is largely unknown. Astrocytes exert a powerful influence on synaptic development, secreting factors to either promote or inhibit synapse formation and neuronal maturation in the developing CNS. Here, we investigated the effects of the partial µ-opioid receptor agonist buprenorphine on astrocyte synaptogenic signaling and morphological development in cortical cell culture. Acute buprenorphine treatment had no effect on the excitatory synapse number in astrocyte-free neuron cultures. In conditions where neurons shared culture media with astrocytes, buprenorphine attenuated the synaptogenic capabilities of astrocyte-secreted factors. Neurons cultured from drug-naïve mice showed no change in synapses when treated with factors secreted by astrocytes from POE mice. However, this same treatment was synaptogenic when applied to neurons from POE mice, indicating a complex neuroadaptive response in the event of impaired astrocyte signaling. In addition to promoting morphological and connectivity changes in neurons, POE exerted a strong influence on astrocyte development, disrupting their structural maturation and promoting the accumulation of lipid droplets (LDs), suggestive of a maladaptive stress response in the developing CNS.

## 1. Introduction

Increased rates of opioid use in recent decades have led to a public health crisis in many countries, especially the United States and Canada. One particularly vulnerable and significantly impacted group is pregnant women, whose opioid use during pregnancy (whether prescription or off-label) can lead to the development of withdrawal symptoms in their children shortly after birth. These symptoms, collectively referred to as neonatal abstinence syndrome (NAS, sometimes more specifically referred to as neonatal opioid withdrawal syndrome) [1], include neurological, respiratory, and gastrointestinal issues that require prolonged hospitalization and, in many cases, pharmacological intervention [2]. Despite the alarming fact that the incidence of NAS has increased nearly six-fold in the past two decades [3,4], relatively little is known about whether prenatal opioid exposure (POE) has long-term consequences for the health and function of the central nervous system (CNS) [5,6].

The few longitudinal studies that have investigated brain development in individuals born with POE and NAS indicate substantial deficits in processes such as movement, attention, memory, cognition, language, and behavior, often persisting years after their initial diagnosis [7,8]. Long-term synaptic dysfunction thus represents a significant maladaptive finding in the brains of individuals exposed to opioids in utero. We previously used a mouse model of maternal drug exposure to observe region-specific changes in synaptic connectivity that persist well after the cessation of drug administration [9]. Intriguingly, these altered patterns of connectivity could be further modulated by the manipulation of signaling pathways associated with astrocytes. In recent years, the formation and maturation of synaptic networks in the brain has been shown to be highly regulated by astrocytes [10], which can either promote or inhibit the process of synaptogenesis via both contact-mediated interactions and secreted factors [11]. In vitro neuronal culture systems have been used in the identification of a number of these secreted factors, including thrombospondin (TSP) [12], secreted protein acidic and rich in cysteine (SPARC) and SPARC-like 1 (i.e., hevin) [13], glypican (Gpc) [14], and chordin-like 1 (Chrdl1) [15]. Furthermore, not only have astrocytes been revealed to be critical for the formation and function of synaptic networks, but the relationship has also been found to be highly reciprocal, with neuronal contact and signaling driving the development and maturation of astrocytes [16,17,18]. It is unsurprising, then, that astrocyte dysfunction and impaired astrocyte–neuron interactions have previously been implicated in the neurobiology of substance use and the pathogenesis of addiction [19,20].

Opioids have been shown to impact astrocyte biology by modulating the expression and secretion of several of the previously mentioned synaptogenic factors [21,22] and other gliotransmitters [23], as well as exerting a strong influence on the differentiation and maturation of these cells [24], though rarely in the context of prenatal exposure. In this study, we investigated the impact of prenatal exposure to the opioid buprenorphine on the excitatory synaptogenic signaling and morphological development of astrocytes during a key time period for these processes, with potential implications for long-term synaptic connectivity and function in the POE brain.

## 2. Materials and Methods

### 2.1. Animals

All animal procedures were performed in accordance with Marshall University’s Institutional Animal Care and Use Committee (IACUC) regulations (W.C.R. protocols 696/697/698). Adult Sprague Dawley rat mothers and male-only newborn litters, as well as timed pregnant C57Bl/6J female mice (arriving on gestational day 6), were obtained from Hilltop Lab Animals (Scottdale, PA, USA). All animals were given ad libitum food and water. All efforts were made to minimize animal suffering and the total number of animals used.

### 2.2. Prenatal Drug Treatment

Prenatal drug treatment was performed according to our previously established protocol [9]. Briefly, on embryonic day 6 (E6), pregnant C57Bl/6J females were given access to 1 mL of a 1:1 sweetened condensed milk/water solution served in a plastic 35 mm dish. Starting on E7, pregnant females were given free access to a once-daily 1 mL solution of either 1:1 condensed milk/water containing pharmaceutical-grade buprenorphine hydrochloride (CIII) (5 mg/kg; Spectrum Chemical, Gardena, CA, USA) or vehicle control. Drug doses were calculated based on the weight of pregnant females on E7. The oral route of buprenorphine administration mimics that of humans, while the dosage represents the minimum effective dosage to achieve analgesia in a rodent model [25]. Daily dosing continued through the birth of pups. Mice were routinely observed to consume the entirety of the milk/water solution. We previously showed this paradigm to be sufficient to achieve brain tissue concentrations of buprenorphine in the pups that were within the clinically relevant range of infants born with NAS [9,26,27].

### 2.3. Cortical Neuron Purification, Culture, and Treatment

Cortical neurons were purified separately from either male postnatal day 1 (P1) Sprague Dawley rats (no prenatal drug exposure) or mixed-sex P1 C57Bl/6J mice (prenatally exposed to either buprenorphine or vehicle control) by sequential immunopanning following established protocols [28,29]. Briefly, following rapid decapitation, brains were removed from the skull and cortices were dissected. Following 45 min enzymatic digestion with papain (~7.5 units/mL; Worthington, Lakewood, NJ, USA) supplemented with DNase (Worthington) dissolved in Dulbecco’s phosphate-buffered saline (DPBS; Gibco, Waltham, MA, USA) at 34 °C, the cell solution was subjected to sequential low/high concentrations of ovomucoid (Worthington) to gradually neutralize papain. The cell solution was then passed through a 20 µm Nitex mesh filter (Sefar, Buffalo, NY, USA) prior to a series of negative immunopanning steps to remove unwanted cells and debris: Petri dishes were coated with Bandeiraea Simplicifolia Lectin I (Vector Laboratories, Burlingame, CA, USA), followed by AffiniPure goat-anti mouse IgG + IgM (H + L) (Jackson Immuno Research Labs, West Grove, PA, USA) and AffiniPure goat-anti rat IgG + IgM (H + L) (Jackson Immuno Research Labs) antibodies. To further purify neurons (>95%), the cell solution was subjected to positive immunopanning by passaging onto Petri dishes coated with primary antibodies against neural cell adhesion molecule L1 (for rat neurons, mouse anti-rat ASCS4; Developmental Studies Hybridoma Bank, Iowa City, IA, USA; for mouse neurons, rat anti-mouse L1; MilliporeSigma, Burlington, MA, USA). After final washes, centrifugation, and resuspension of the cells in serum-free neuronal growth medium (NGM), neurons were plated at a density of 60 K/well on poly-D-lysine (PDL; MilliporeSigma) and laminin (R&D Systems, Minneapolis, MN, USA)-coated coverslips in a 24-well plate. Serum-free NGM consisted of Neurobasal (Gibco), B27 supplement (Gibco), 2 mM GlutaMax (Gibco), 100 U/mL penicillin/streptomycin (Gibco), 1 mM sodium pyruvate (Gibco), 50 ng/mL brain-derived neurotrophic factor (BDNF; Peprotech, Cranbury, NJ, USA), 20 ng/mL ciliary neurotrophic factor (Peprotech), 4.2 µg/mL forskolin (MilliporeSigma), 2 µM cytosine arabinoside (AraC; MilliporeSigma) to halt contaminating astrocyte proliferation (rat neurons only), and 100 µg/mL primocin (Invivogen, San Diego, CA, USA) (mouse neurons only).

For rat neurons, after 2 days in vitro (DIV2) at 37 °C/5% CO_2_, half of the NGM in each well was replaced with fresh, equilibrated NGM of identical composition with the exception of Neurobasal Plus (Gibco) instead of Neurobasal and GlutaMax, B27 Plus (Gibco) instead of B27, and no AraC; this media was then used for feedings every 2–3 days for the duration of the experiment prior to fixation for synaptic immunocytochemistry (ICC) on DIV13. For some treatment groups, buprenorphine hydrochloride (500 nM) and/or gabapentin (32 µM; Spectrum Chemical) were added to NGM for feedings performed on DIVs 8 and 11.

For mouse neurons, on the morning of DIV2, half of the NGM in each well was replaced with fresh, equilibrated NGM of identical composition to the DIV0 plating media, but with the addition of 2 µM AraC. On DIV3, AraC was removed via complete NGM media change (otherwise identical to DIV0 plating media).

### 2.4. Cortical Astrocyte Purification, Culture, Transfection, and Lysis

Cortical astrocytes were purified separately from either male P1 Sprague Dawley rats (no prenatal drug exposure) or mixed-sex P1 C57Bl/6J mice (prenatally exposed to either buprenorphine or vehicle control) following a similar protocol to neurons, as described above [29]. After the Nitex mesh filtering step, the cell solution was centrifuged and the pellet resuspended in astrocyte growth media [AGM; Dulbecco’s modified Eagle medium +GlutaMax (Gibco), 10% heat-inactivated fetal bovine serum (MilliporeSigma), 10 µM hydrocortisone (MilliporeSigma), 100 U/mL penicillin/streptomycin, 5 µg/mL insulin (Sigma), 1 mM sodium pyruvate, 5 µg/mL N-acetyl-L-cysteine (MilliporeSigma)]. Then, 15–20 million cells were plated on PDL-coated 75 mm^2^ flasks and incubated at 37 °C/5% CO_2_. On DIV3, AGM was removed and replaced with DPBS. To isolate the adherent monolayer of astrocytes, flasks were washed 2–3× with DPBS and then shaken vigorously by hand 3–6× for 15 s each. DPBS was then replaced with fresh AGM. AraC was added to the AGM from DIVs 5–7 to minimize astrocyte proliferation. On DIV7, astrocytes were passaged into transwell inserts (Corning Life Sciences, Corning, NY, USA) at a density of 125 K per insert (for treatment of neurons) at either 400 K per well of a 6-well culture plate (for transfection and co-culture with neurons) or 600 K per well of a 6-well culture plate (for lysis and Western blot analysis). On the morning of DIV8, astrocyte inserts were added to some wells of the neuron plates (for synaptogenic signaling/neuron Sholl analysis experiments).

For astrocyte transfection (DIV8), Lipofectamine LTX with Plus Reagent (Invitrogen, Carlsbad, CA, USA) was used per manufacturer’s protocols. Briefly, 2 µg of pAAV-CAG-tdTomato was used to transfect each well of the astrocytes, incubating for 3 h at 37 °C/5% CO_2_. pAAV-CAG-tdTomato (codon diversified) was as gift from Edward Boyden (Addgene Plasmid #59462; http://n2t.net/addgene:59462 (accessed on 9 May 2024); RRID:Addgene_59462). On DIV10, transfected astrocytes were passaged and plated in NGM at 20 K/well directly on top of neurons. Cells were cultured together in this way for 48 h prior to fixation and staining, as in [17].

For astrocyte lysis (DIV10), the plate containing the astrocytes was removed from the incubator and placed on ice. AGM was aspirated and cells were washed twice with 1 × Tris-buffered saline + 1 mM CaCl_2_ + 2 mM MgCl_2_ and then incubated on ice for 20 min with lysis buffer (25 mM Tris, 150 mM NaCl, 1 mM CaCl_2_, 1 mM MgCl_2_, 0.5% NP-40 [Thermo Scientific, Waltham, MA, USA], EDTA-free protease inhibitor [Roche, Basel, Switzerland]). Lysed astrocytes were collected into 1.5 mL microcentrifuge tubes, vortexed, and then centrifuged at 15,000 rpm for 5 min at 4 °C. The supernatant was transferred to a new 1.5 mL tube and stored at −80 °C until needed for protein analysis by Western blot.

### 2.5. Immunocytochemistry (ICC)

DIV12–13 (mouse) or DIV14 (rat) cells in 24-well plates were fixed and stained for ICC according to established protocols [28,29]. Briefly, growth media was aspirated and replaced with warm 4% paraformaldehyde (PFA) (Electron Microscopy Sciences, Hatfield, PA, USA) in phosphate-buffered saline (PBS) for 7 min. PFA was removed and cells were washed 3× with PBS. For synaptic ICC, cells were then exposed to blocking buffer containing 0.2% Triton X-100 (Roche) in 50% normal goat serum (NGS; Jackson Immuno Research)/50% antibody buffer (PBS containing 1% bovine serum albumin [BSA; MilliporeSigma], 0.04% NaN_3_ [MilliporeSigma], 0.2% Triton X-100) at room temperature for 30 min. Cells were then washed 3× with PBS and treated with 10% NGS/90% antibody buffer containing primary antibodies against Bassoon (1:500; clone SAP7F407 mouse; Enzo/Assay Designs ADI-VAM-PS003, Farmingdale, NY, USA) plus Homer1 (1:500; rabbit; GeneTex GTX103278, Irvine, CA, USA) or FABP7 (1:2000; rabbit; Neuromics RA22137, Edina, MN, USA) at 4 °C overnight in the dark. The following morning, cells were washed 3× with PBS, followed by 2 h at room temperature in 10% NGS/90% antibody buffer containing the following fluorescently conjugated secondary antibodies: goat anti-mouse AlexaFluor 488 (1:500; Invitrogen) plus goat anti-rabbit AlexaFluor 594 (1:500; Invitrogen) or goat anti-rabbit 488 (1:500; Invitrogen). After a final round of 3× PBS washes, coverslips were transferred to glass slides with Vectashield mounting medium containing 4′,6-diamidino-2-phenylindole (DAPI; Vector Laboratories), sealed with clear nail polish, and imaged on a Leica DM5500B (Wetzlar, Germany) fluorescence microscope with a 63×/1.4 NA objective at 1920 × 1440 resolution. The DAPI channel was not imaged but instead was used to select non-overlapping cells for imaging in a non-biased manner. In a separate series of experiments, immediately following PFA fixation, cells were incubated at room temperature for 30 min in PBS containing LipidSpot reagent (1:1000; Biotium, Fremont, CA, USA), and then washed 3× with PBS and mounted onto glass slides with Vectashield + DAPI.

### 2.6. Cell Viability Assay

Cell viability was determined on DIV10 (astrocytes) or DIV12 (neurons) with the LIVE/DEAD Viability/Cytotoxicity Kit (Invitrogen) according to manufacturer’s protocols. Cells were imaged on a Nikon Eclipse TS100 inverted fluorescence microscope (Melville, NY, USA) with a 10×/0.25 NA objective at 1440 × 1024 resolution. Live cells were stained with Calcein-AM, while dead cells were stained with ethidium homodimer-1.

### 2.7. Western Blotting

Western blotting was performed on cortical astrocyte lysates prepared on DIV10. Lysate protein concentrations were determined using the Micro BCA Protein Assay Kit (Pierce Protein Biology, Rockford, IL, USA) according to manufacturer’s protocols. Briefly, 30 µg of protein was loaded into each well of a 4–15% Stain-Free polyacrylamide gel (Bio-Rad, Hercules, CA, USA) and resolved via SDS-PAGE. Each gel was transferred to a methanol-activated Immobilon-FL PVDF membrane (MilliporeSigma) which was then UV-activated and imaged for total protein content on a Chemidoc MP (Bio-Rad). Membranes were blocked at room temperature for 1 h in a 1:1 mixture of PBS and fluorescent-blocking buffer (Rockland Immunochemicals, Limerick, PA, USA), followed by an overnight incubation at 4 °C in blocking buffer plus 0.1% Tween 20 (Caisson Labs, Smithfield, UT, USA) along with one of the following primary antibodies: rabbit anti-Chrdl1 (1:250; MilliporeSigma HPA000250), rabbit anti-FABP7 (1:2000; Neuromics RA22137), rabbit anti-Gpc4 (1:500; Proteintech 13048-1-AP, Rosemont, IL, USA), mouse anti-Gpc6 (1:2000; R&D Systems AF1053, Minneapolis, MN, USA), mouse anti-hevin (1:2000; R&D Systems AF2836), mouse anti-SPARC (1:1000; R&D Systems AF942), goat anti-TSP1 (1:1000; R&D Systems AF3074), goat anti-TSP2 (1:1000; R&D Systems AF1635), or goat anti-TSP4 (1:1000; R&D Systems AF2390). The next day, membranes were incubated at room temperature for 1 h in blocking buffer plus 0.1% Tween 20 along with species-appropriate fluorescently labeled secondary antibodies (Multifluor Red [goat anti-rabbit] or Green [goat anti-mouse], 1:1000; ProteinSimple, Minneapolis, MN, USA; donkey anti-goat Alexa Fluor 488, 1:1000; Invitrogen). Fluorescent antibody detection was performed with the Chemidoc MP. Western blot band intensity was quantified with Image Lab (Bio-Rad), using the previously acquired total protein images (Appendix A) for lane normalization. Several membranes were stripped once with 1 M NaOH for 15 min followed by 4 × 5 min washes with PBS + Tween before being re-probed with a different primary antibody.

### 2.8. Image Analysis and Statistics

To minimize bias, all experimental samples were assigned random blinding codes prior to imaging and analysis.

The synapse quantification of ICC-stained images was performed by a trained analyst using the ImageJ 1.29 (NIH, Bethesda, MD, USA) custom Puncta Analyzer plugin (C. Eroglu, Duke University) [28,29]. The plugin allows for the rapid counting of pre- (i.e., Bassoon), post- (i.e., Homer1), and co-localized synaptic puncta, determined by user-defined thresholds for each individual channel. This approach provides an accurate estimation of the synapse number based on the precise localization of pre- and postsynaptic proteins, which localize to separate compartments of neurons and only appear to overlap in fluorescence-based ICC when directly opposed at synaptic junctions [30]. Briefly, a user-defined circular region of interest surrounding the cell, approximately 3 cell diameters wide, is divided into separate channels. After background subtraction (rolling ball radius = 50), the analyst sets the intensity threshold for each channel with the goal of isolating the “true” signal (i.e., puncta; minimum pixel size = 4) from the background. The plugin then automatically calculates the number of individual puncta for each channel, as well as the number of co-localized puncta, which we categorize as synapses.

Neuronal and astrocytic morphometrics were quantified via Sholl analysis using a plugin available for FIJI/ImageJ [31] using a previously published protocol [16,17]. This method provides an estimate of branching complexity by counting the number of times the fluorescent cellular signal intersects with each iteration of a series of concentric circles drawn at fixed distances emanating from the cell soma. Briefly, either the postsynaptic Homer1 label (for neurons) or the tdTomato cell fill (for astrocytes) was color-thresholded in FIJI (2.9.0) to create a mask that captured the morphology of the soma plus processes for each imaged cell. The Sholl analysis plugin, freely available from the Neuroanatomy/SNT repository for FIJI [32], was then used to calculate intersections of the mask channel along a series of concentric circles spaced 1 µm apart beginning at radius 10 µm (to account for the soma).

The same tdTomato color-thresholding/masking process in FIJI described above was used to isolate individual, labeled astrocytes in the astrocyte/neuron co-cultures prior to calculating the mean fluorescence intensity for the anti-FABP7-stained cells, as well as for lipid droplet (LD) counts for the LipidSpot-stained cells. The portion of the LipidSpot channel that overlapped with the astrocyte mask was processed with the ilastik machine learning tool [33]. Ilastik’s “pixel classification” function was used to analyze the LipidSpot channel and to separate the LD signal from the background, which was then saved to a new image. This image was then exported to FIJI, where the “analyze particles” function was used to calculate the number and area of individual LDs within the boundaries of each labeled astrocyte.

Raw data were captured in Microsoft Excel spreadsheets (version 2404, Redmond, WA, USA), and statistical analysis was performed with Graphpad Prism (San Diego, CA, USA) or with the statsmodels module for Python (for analysis of co-variance/ANCOVA with Tukey’s post hoc testing for Sholl analysis). All data are presented as mean ± standard error of the mean. The Kolmogorov–Smirnov test was used to test distributions for normality. When data were distributed normally, parametric tests including an unpaired *t*-test and ANOVA were performed. In the event of non-normal distributions, the nonparametric Mann–Whitney or Kruskal–Wallis tests with post hoc Dunn’s multiple comparisons test were performed to detect significant differences in means between groups (*p* < 0.05). The most relevant statistical analyses are presented in Section 3, figures, and figure captions; a comprehensive list of all analyses can be found in Appendix A.

## 3. Results

### 3.1. Acute Buprenorphine Exposure Inhibits Astrocyte-Mediated Synaptogenesis

To determine the effect of acute opioid exposure on synapse formation during the peak synaptogenic window for rodent CNS [34,35], we purified neurons and astrocytes from the cortices of male P0-1 rat pups (Figure 1A). Only male pups were used, due to our previous finding that, at least in rats, astrocyte-mediated synaptic development demonstrates sex differences, with male-derived neurons showing a significantly greater response to astrocyte-secreted factors than those from females [29]. Neurons were cultured in either the presence or absence of astrocytes in inserts (allowing the cells to share media and thus secreted factors), followed by acute treatment with buprenorphine. As a control, we also used the drug gabapentin, which was previously shown to interfere with astrocyte synaptogenic signaling in a calcium-channel-independent manner [36]. ICC followed by fluorescence microscopy was used to quantify the co-localization of the presynaptic-specific protein bassoon with postsynaptic-specific homer1, which indicated the location of excitatory synapses (Figure 1B). We observed a significant main effect between treatment groups (H = 24.71, *p* = 0.0009). In the absence of astrocytes, cortical neurons did not demonstrate any changes in bassoon/homer1 puncta co-localization following acute buprenorphine and/or gabapentin exposure (Figure 1B,C). As expected, neurons cultured together with an astrocyte insert showed increased puncta co-localization (and therefore increased synapse numbers) compared to neurons cultured alone (neurons + astro: 65.48 ± 6.89, neurons only: 32.95 ± 3.11; *p* = 0.0141, Dunn’s multiple comparisons test; Figure 1B,C); however, this was only observed in the conditions that were unexposed to either drug. The addition of buprenorphine or gabapentin, either alone or in combination, abolished the synaptogenic effect of the astrocytes (Figure 1B,C). These results indicate that, similarly to gabapentin, acute exposure to opioids attenuates the synapse-promoting properties of astrocyte-secreted factors during early postnatal development.

### 3.2. Prenatal Buprenorphine Promotes Neuroadaptive Changes in Response to Astrocyte-Secreted Factors

To test whether astrocyte-mediated synaptogenesis is impaired after prolonged prenatal exposure to opioids, pregnant mouse dams were given daily access to either vehicle control or buprenorphine (5 mg/kg) in a 1 mL condensed milk mixture from E7 until the birth of their litter. This dosing paradigm models the oral route of administration typical for buprenorphine use in humans [37] and is timed to coincide with a period of rapid synaptogenesis and astrocyte maturation in the rodent brain [16,34]. We have previously confirmed that this dosing strategy results in clinically relevant brain concentrations in newborns [9]. Once the litters were born, P0-1 male and female pups were euthanized (for feasibility purposes, sexes had to be combined due to small litter sizes), and cortical neurons and astrocytes were purified separately. Neurons from either the vehicle or buprenorphine-exposed (i.e., POE) pups were then cultured by themselves or together with astrocyte inserts in several different combinations: in some wells, neurons and paired astrocyte inserts came from mice with the same prenatal treatment, while in others the treatments were mismatched (e.g., astrocytes from POE mice were inserted into wells with neurons from vehicle mice, while POE neurons were cultured with vehicle-exposed astrocyte inserts). After 2 weeks of culture, neurons were then fixed, stained for bassoon/homer1 ICC, and imaged (Figure 2A). The quantification of co-localized synaptic puncta (Figure 2B) revealed a significant main effect between treatments (H = 77.66, *p* < 0.0001). In our positive control comparison, we confirmed that vehicle neurons cultured with vehicle astrocyte inserts showed a significant increase in the number of co-localized synaptic puncta compared to vehicle neurons that only received growth media (No Astro: 7.83 ± 1.30, Veh Astro: 21.24 ± 2.473; *p* < 0.0001, Dunn’s multiple comparisons test; Figure 2A,B). However, when vehicle neurons were exposed to secreted factors from POE astrocytes, no changes in synapse number compared to the No Astro control were observed (POE Astro: 6.88 ± 0.82; *p* > 0.9999, Dunn’s multiple comparisons test; Figure 2A,B). This finding suggests that POE had interfered with the developmental secretome of astrocytes [38,39]. By contrast, neurons from buprenorphine-exposed brains demonstrated a significant increase in synapse number when cultured with astrocyte inserts (derived from either vehicle or POE brains) compared to growth media alone (No Astro: 8.13 ± 1.55; Veh Astro: 20.14 ± 1.97, *p* < 0.0001, POE Astro: 15.94 ± 2.13, *p* = 0.0391; Dunn’s multiple comparisons test; Figure 2A,B). Taken together, these results show that astrocytes from mice prenatally exposed to buprenorphine were still capable of promoting synaptogenesis via secreted factors, but only when neurons were from pups that had experienced the same prolonged opioid exposure.

We next investigated whether the observed changes in synapse number were merely due to adaptations in the overall morphology of the neurons. We thresholded the previously imaged postsynaptic homer1 channels (Figure 2A) in a way that enabled us to create a mask outlining the basic morphology of each cell, including its soma and dendrites. Sholl analysis of the neuronal masks revealed that there were no differences in branching complexity between vehicle and POE neurons cultured in the absence of astrocytes (Veh Neuron/No Astro vs. POE Neuron/No Astro: *p* > 0.9999; ANCOVA with pairwise comparisons plus Bonferroni correction; Figure 2C). Interestingly, vehicle astrocyte inserts induced a similar degree of neuronal branching complexity regardless of whether the neurons were from vehicle or POE mice (Veh Neuron/No Astro vs. Veh Neuron/Veh Astro: *p* = 0.0011; POE Neuron/No Astro vs. POE Neuron/Veh Astro: *p* < 0.0001; Veh Neuron/Veh Astro vs. POE Neuron/Veh Astro: *p* > 0.9999; ANCOVA with pairwise comparisons plus Bonferroni correction; Figure 2C). However, astrocytes from mice previously exposed to buprenorphine showed a strongly inhibitory effect on neuronal branching; this was the case for both vehicle and POE neurons (Veh Neuron/No Astro vs. Veh Neuron/POE Astro: *p* < 0.0001; POE Neuron/No Astro vs. POE Neuron/POE Astro: *p* < 0.0001; ANCOVA with pairwise comparisons plus Bonferroni correction; Figure 2C). The regulation of neuronal morphological development by astrocyte-secreted factors is therefore both highly vulnerable to the effects of prenatal opioids as well as being separate from their effects on synapse number, although we cannot completely rule out the possibility that decreased dendritic complexity may induce limitations on the synaptogenic potential in these cultures.

### 3.3. POE Does Not Affect the Short-Term Survival of Cultured Neurons and Astrocytes

Having confirmed that POE can detrimentally affect synapse number and cellular morphology, we then wanted to rule out the possibility that these effects were merely due to differences in cell viability or proliferation in our cultures. Cell viability assays conducted on both neuron-only and astrocyte-only cultures confirmed that POE treatment had no effect on the survival of neurons (Vehicle: 29.35 ± 1.06%, POE: 30.21 ± 0.97%; *p* = 0.5553, unpaired *t*-test; Figure 3A,B) or astrocytes (Vehicle: 94.37 ± 0.65%, POE: 94.83 ± 0.34%; *p* = 0.8063, Mann–Whitney test; Figure 3C,D) at this early stage of postnatal development. Furthermore, there were no effects of POE on astrocyte proliferation, as the total number of live cells was unchanged from the vehicle control (Vehicle: 197.9 ± 12.99, POE: 195.7 ± 7.43; *p* = 0.8841, unpaired *t*-test; Figure 3E).

### 3.4. Buprenorphine Alters the Expression of Astrocytic Proteins Known to Regulate Synaptic and Neuronal Maturation

With the knowledge that POE disrupts astrocyte-to-neuron signaling, as evidenced by changes in synapse number and neuronal branching (Figure 2), we then wanted to investigate potential mechanisms underlying these adaptations. Using vehicle and POE astrocyte lysates, we performed a Western blot analysis of several secreted proteins with known roles in regulating synaptic development and neuronal morphology and maturation (Figure 4A). Compared to vehicle astrocytes, POE astrocytes had higher levels of TSP1 and TSP4 as well as a lower expression of SPARC (TSP1: 142.3 ± 9.0% compared to Veh, *p* = 0.0308; TSP4: 204.4 ± 15.4% compared to Veh, *p* = 0.0023; SPARC: 70.2 ± 10.4%compared to Veh, *p* = 0.0405; unpaired *t*-test; Figure 4B). We also measured the expression of fatty acid binding protein-7 (FABP7), which is highly expressed by juvenile and mature astrocytes and is known to regulate neuronal branching and synapse number/function [40]. Intriguingly, we found a marked upregulation of FABP7 in the POE astrocytes compared to the vehicle (201.8 ± 29.0% compared to Veh, *p* = 0.0220; unpaired *t*-test; Figure 4B). Taken together, these results reveal that POE induces widespread changes in the expression of astrocytic factors that may have profound effects on the development of neural connectivity.

### 3.5. Disruptions in Astrocyte Morphology and Lipid Droplets Associated with POE

After observing disrupted neuronal morphology following POE (Figure 2), as well as changes in astrocytic protein expression (Figure 4), we next asked whether astrocyte morphology was similarly impacted in our culture model. Because astrocytes have previously been shown to develop more complex morphologies when cultured on top of neurons [16], we took tdTomato-expressing astrocytes from Veh or POE mice and passaged them directly onto a monolayer of Veh or POE neurons (Figure 5A). Then, 48 h later, differences in astrocyte morphology were readily apparent, with Veh astrocytes grown on Veh neurons showing the highest degree of complexity (Figure 5A,B). When astrocytes isolated from POE mice were in contact with Veh neurons, branching complexity was significantly decreased compared to the Veh/Veh group (*p* < 0.0001, ANCOVA with pairwise comparisons plus Bonferroni correction; Figure 5A,B). However, the least complex morphologies were seen when astrocytes were grown on top of POE neurons, with no statistical difference between the Veh or POE astrocyte groups (Veh Neuron/Veh Astro vs. POE Neuron/Veh Astro: *p* < 0.0001; Veh Neuron/Veh Astro vs. POE Neuron/POE Astro: *p* < 0.0001; POE Neuron/Veh Astro vs. POE Neuron/POE Astro: *p* > 0.9999; ANCOVA with pairwise comparisons plus Bonferroni correction; Figure 5A,B). This result indicates that the greatest limiter of astrocyte structural elaboration in this scenario is prior opioid exposure of the neurons with which they are in contact.

FABP7, which was upregulated in our astrocyte-only POE cultures (Figure 4B), regulates neuronal morphology [40], but whether it has a similar role in astrocyte development has yet to be elucidated. IHC staining for FABP7 revealed an increased expression in POE astrocytes, but this increase was similar for cells grown on both Veh and POE neurons (Veh Neuron/Veh Astro vs. Veh Neuron/POE Astro: *p* = 0.0015; POE Neuron/Veh Astro vs. POE Neuron/POE Astro: *p* = 0.0016; F = 10.39, one-way ANOVA with Tukey’s multiple comparisons test; Figure 5C,D). As cellular chaperones for lipids and their component molecules, such as polyunsaturated fatty acids (PUFAs), FABP7 plays a key role in the intracellular accumulation of LDs, especially under conditions of stress [41,42]. To investigate LD accumulation in astrocytes, we used a cellular stain for LDs (Figure 5E) and observed a robust upregulation in LD number (Veh Neuron/Veh Astro vs. Veh Neuron/POE Astro: *p* = 0.0003; Veh Neuron/Veh Astro vs. POE Neuron/Veh Astro: *p* < 0.0001; Veh Neuron/Veh Astro vs. POE Neuron/POE Astro: *p* < 0.0001; H = 43.42, Kruskal–Wallis with Dunn’s multiple comparisons test; Figure 5F) and area (Veh Neuron/Veh Astro vs. Veh Neuron/POE Astro: *p* = 0.0030; Veh Neuron/Veh Astro vs. POE Neuron/Veh Astro: *p* < 0.0001; Veh Neuron/Veh Astro vs. POE Neuron/POE Astro: *p* = 0.0014; H = 25.65, Kruskal–Wallis with Dunn’s multiple comparisons test; Figure 5G) when either neurons or astrocytes were from POE mice. After accounting for cell area, the total increase in intracellular LD accumulation was highest when neurons were from POE mice (Veh Neuron/Veh Astro vs. Veh Neuron/POE Astro: *p* < 0.0001; Veh Neuron/Veh Astro vs. POE Neuron/Veh Astro: *p* < 0.0001; Veh Neuron/Veh Astro vs. POE Neuron/POE Astro: *p* < 0.0001; Veh Neuron/POE Astro vs. POE Neuron/POE Astro: *p* = 0.0160; H = 70.38, Kruskal–Wallis with Dunn’s multiple comparisons test; Figure 5H).

Overall, these results indicate that (1) astrocytes show increased LD accumulation when previously stressed by prenatal opioids; (2) that this stress is exacerbated when physically contacting neurons that were exposed to the same stress; and (3) that this accumulation may be linked to changes in the astrocytic expression of FABP7.

## 4. Discussion

Much of what we know about the effects of opioids on the brain comes from studies in either adolescents or adults, i.e., long after the maturation of astrocytes and establishment of synaptic circuitry. As such, investigations into the effects of prolonged opioid exposure on developing neurons and glia are limited [24]. We previously showed that excitatory synaptic connectivity is significantly disrupted in the brains of juvenile mice with early life exposure to buprenorphine, and that the extent of this disruption is impacted by modulating astrocyte synaptogenic signaling pathways [9]. To begin to elucidate a cellular mechanism by which these synaptic impairments may occur, we have turned to an in vitro culture system to study cell-type-specific effects of prenatal opioids on CNS development. We performed our cell isolations at P1, a time at which neurogenesis has already occurred but when astrocytes are only starting to populate the rodent cortex. Thus, our prenatal exposure paradigm would primarily affect the opioid receptor-expressing neural stem cells that later differentiate into astrocytes around the time of purification. We then studied in vitro development over the first 1–2 postnatal weeks. In rodents, this period, roughly equivalent to human fetal development in the mid-second trimester [43], overlaps with a window of rapid synapse formation and maturation that lays the architectural foundation for the neural circuits that will populate the mature brain [34]. Acutely administered buprenorphine did not have a direct effect on excitatory synapse formation in our neuron-only cultures; rather, it attenuated the synaptogenic properties of astrocytes that shared media with the neurons (Figure 1). Like neurons, astrocytes express opioid receptors on their surface [44] that, when activated, lead to increases in intracellular calcium and subsequent changes in gliotransmission [23,45]. Previously, in vitro treatment with morphine, an opioid that is most active at the µ-opioid receptor (MOR), was shown to promote changes in the astrocytic expression of TSPs 1 and 2, concomitant with aberrant neurite outgrowth and excitatory synapse number [21,22]. MOR activation was also shown to inhibit the growth of cultured astrocytes through the suppression of DNA synthesis [46], although this study did not determine whether astrocyte-neuronal communication was subsequently impaired. Though we saw no effects on astrocyte proliferation or survival with POE (Figure 3), we did confirm changes in the expression of multiple TSPs as well as SPARC; these proteins, in addition to having known effects on synapse formation and maturation, are critical components of the extracellular matrix (ECM). The remodeling of the ECM has been linked to the modulation of synaptic transmission and plasticity in a manner facilitating opioid relapse in adults [47], raising the possibility that opioid-induced disruptions in these proteins during early development may have significant lasting effects on astrocyte-regulated synaptic connectivity. Disruptions in astrocytic signaling were also implicated in the impaired synaptic plasticity observed following concomitant treatment with astrocyte-conditioned media and morphine in the nucleus accumbens [48]. A decreased expression of BDNF, a key regulator of synaptic connectivity and maturation that can be synthesized and released by astrocytes, was observed in a rat model of prenatal buprenorphine exposure [49]. Another aspect of astrocytic signaling not directly addressed here, but which is nonetheless highly relevant to synaptic health, is the release of cytokines. Opioid exposure increases proinflammatory cytokine secretion by astrocytes [50], which may in turn affect the downstream production and/or release of synapse-mediating factors such as hevin and SPARC. In addition to regulating excitatory glutamatergic synaptogenesis, astrocytes have also been shown to promote the formation of inhibitory GABAergic synapses [51,52]; these possibilities are currently being investigated.

Repeated exposure to opioids is known to result in significant adaptations to the synaptic circuitry that manifest as impaired neurotransmission, which can persist through the period of withdrawal and beyond [53]. Our results, showing the preservation of synaptogenic signaling by opioid-exposed astrocytes acting on opioid- but not vehicle-treated neurons (Figure 2), suggest a complex neuroadaptive response that somehow maintains synapse-promoting pathways despite drug-induced disruptions in astrocyte-to-neuron signaling. These results are consistent with our previous in vivo findings, showing similar or even increased numbers of excitatory synapses in cortical areas of 21-day-old mice that had early life exposure to buprenorphine [9]. However, different models of POE have shown reductions in neuronal density, such as in the motor cortex following a combinatorial maternal treatment of oxycodone and subsequent methadone maintenance (though this effect was highly region-, layer-, and sex-specific) [54]. Opioids have also been shown to change the expression of apoptotic proteins such as caspase-3, Bcl-2, and Bax in hippocampal neurons [55], although it is unclear whether this was the result of opioids acting directly on neurons or a consequence of the disrupted astrocytic support of neurons. Further experiments are necessary to elucidate the precise nature of this neuroadaptation in response to the opioid-mediated impairment of astrocyte-neuronal communication in early development.

Mei et al. [56] previously demonstrated that dendritic outgrowth was severely reduced in the visual cortex after prenatal exposure to morphine. We observed similar disruptions in neuronal branching complexity in our cultures (Figure 2); however, our approach of culturing vehicle- and opioid-exposed neurons and astrocytes in different combinations led to the finding that it was the actions of opioids on astrocytes that led to the most significant deficits in neuronal morphology. We also found that this relationship was highly reciprocal, as astrocytes grown on top of POE neurons achieved the lowest degree of morphological complexity observed in our culture system (Figure 5). While investigating potential mechanisms linking these opioid-induced disruptions in morphological maturation and synaptic development, we found that FABP7, previously revealed to be a strong driver of neuronal morphogenesis [40], was significantly elevated in the POE astrocytes (Figure 4 and Figure 5). In the CNS, FABP7 (also known as brain lipid binding protein) is most highly expressed by astrocytes, in which they serve as a carrier to allow water-insoluble PUFAs to be internalized and function inside cells [57,58]. It is through this regulation of lipid metabolism that FABP7 has been proposed to facilitate the astrocytic response to cellular stress; stress induces fatty acid peroxidation, increasing the likelihood of activating programmed cell death cascades. As a defense mechanism, astrocytes can degrade the peroxidated lipids via mitochondrial β-oxidation [59,60,61] or sequester them into LDs [62,63]. Islam et al. [42] showed that LD accumulation is significantly reduced in FABP7 knockout astrocytes, concurrent with an increase in reactive oxygen species (ROS) generation and the activation of pro-apoptotic machinery. Conversely, FABP7 overexpression was associated with increased LD accumulation in human glioma cells [42], underscoring the importance of FABP7 for LD formation. We found significant LD accumulation in cultured astrocytes grown together with POE neurons, i.e., the same conditions that had the least complex astrocytic branching (Figure 5). Though much work still needs to be undertaken to confirm and characterize a prolonged cellular stress response after POE, particularly in vivo, our results suggest that this may be a critical mechanism underlying the deficient astrocyte-synaptic development observed in this study.

One caveat in the interpretation of this study lies with our choice of buprenorphine as the experimental opioid of choice. Most previous preclinical investigations into the vulnerability of the developing CNS to opioids used morphine or methadone, which are both full MOR agonists [24]. However, buprenorphine has only partial agonist activity at the MOR and displays antagonistic effects at the δ- and κ-opioid receptors. These unique mechanisms of action mean that our results should be interpreted with the potential differential responses of the various opioid receptor classes in mind. Regardless, we chose buprenorphine for this study because of its prevalence as the preferred prescribed opioid in medication-assisted treatment (MAT) programs for pregnant women [64] due to its having fewer side effects and less abuse liability than morphine or even methadone [65]. As a result, a recent survey of patients revealed that developing fetuses are now much more likely to be exposed to buprenorphine than any other opioids/opiates [66]. Despite the benefits of buprenorphine for the acute treatment of NAS [67], as well as preclinical evidence that shows less severe cognitive impairments with the use of buprenorphine versus morphine and methadone [68,69,70], there remains a concern of negative developmental outcomes. Recently, in utero buprenorphine exposure was shown to promote hyperactivity in the mesolimbic dopamine pathway and sensorimotor gating deficits in adult rats [71]. Furthermore, in humans, early school-age children who were exposed to MAT-prescribed buprenorphine performed significantly worse than a peer group on tests of hyperactivity, impulsivity, memory, and attention [72]. Determining how these deficits develop at the cellular level may allow us to refine clinical buprenorphine usage in order to reap its benefits while preventing the negative developmental consequences that may result from it.

## 5. Conclusions

This study elucidated the effects of prenatal opioid exposure on cellular and synaptic development at key early stages of neural network formation. We found that the opioid buprenorphine interferes with the process of astrocyte-mediated excitatory synapse formation between cortical neurons. Drug-naïve neurons do not show a synaptogenic response to secreted factors from astrocytes isolated from POE brains; however, those same astrocytes are able to promote synapse formation between neurons from drug-exposed animals, suggestive of a neuroadaptive response to altered astrocyte signaling. Both neurons and astrocytes showed stunted morphological development when cultured together after POE, correlating with increased lipid droplet content in astrocytes. These results, taken together with our previous in vivo work showing aberrant synaptic connectivity in the brains of juvenile mice after POE, strongly implicate impaired glial-mediated synaptic development as a significant driver of POE pathology in the CNS.

## Figures and Tables

**Figure 1 cells-13-00837-f001:**
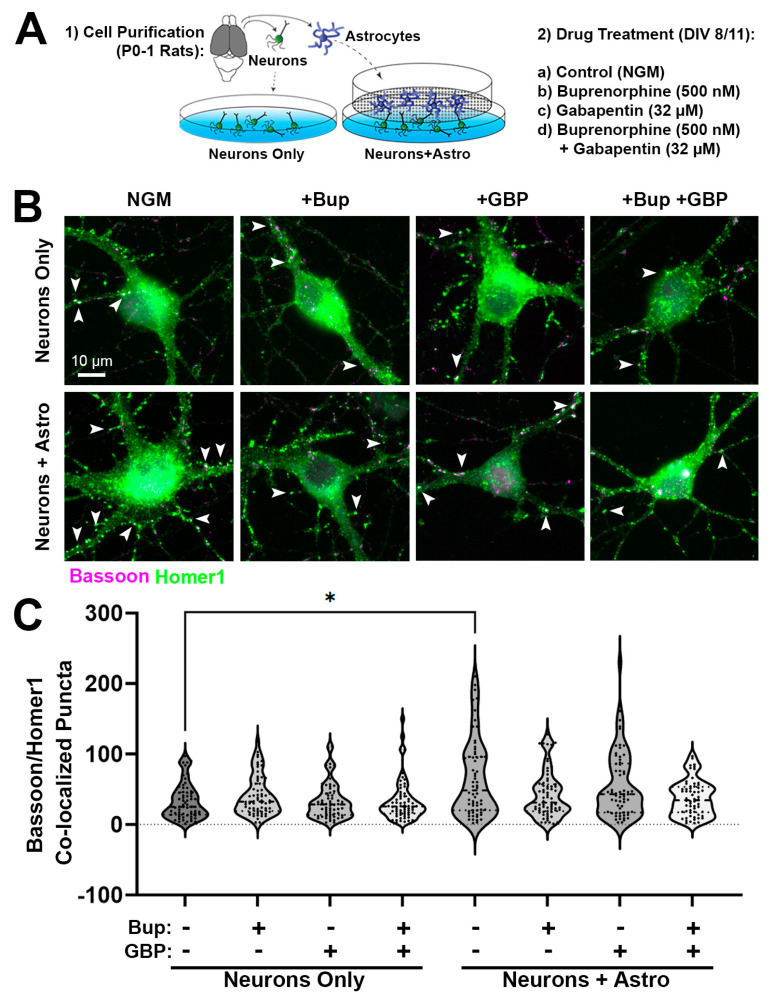
Effects of acute opioid exposure on postnatal synaptic development. (**A**) Primary cell purification and treatment paradigm. (**B**) Rat cortical neurons cultured alone (top row) or with astrocyte inserts (bottom row), with and without various drug treatments (Bup = buprenorphine, GBP = gabapentin). Neurons were labeled with presynaptic (anti-bassoon, magenta) and postsynaptic (anti-homer1, green) primary antibodies. Co-localized puncta, indicating excitatory synapses, appear in white (arrowheads). (**C**) Number of co-localized synaptic puncta at DIV14 within the various treatment groups for neurons only or neurons with astrocyte inserts (*n* = 60–67 total neurons per condition from two independent experimental replicates; * *p* < 0.05, Kruskal–Wallis test with Dunn’s multiple comparisons).

**Figure 2 cells-13-00837-f002:**
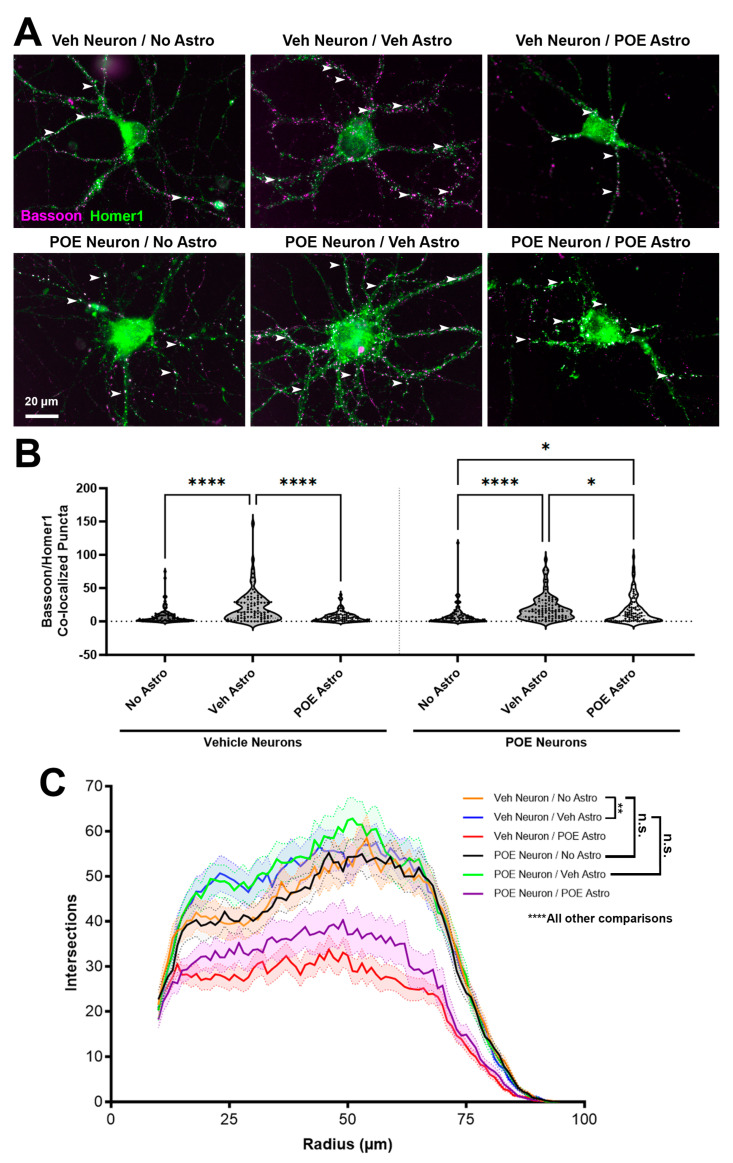
POE alters astrocyte-to-neuron signaling during early development. (**A**) Cortical neurons from vehicle control mice (“Veh neuron”; top row) or from prenatal buprenorphine-exposed mice (“POE neuron”; bottom row). Neurons were cultured alone (“No Astro”) or with astrocyte inserts from vehicle (“Veh Astro”) or buprenorphine-exposed mice (“POE Astro”). Neurons were labeled with presynaptic (anti-bassoon, magenta) and postsynaptic (anti-homer1, green) primary antibodies. Co-localized puncta, indicating excitatory synapses, appear in white (arrowheads). (**B**) Number of co-localized synaptic puncta at DIV13 for vehicle control or POE neurons treated with No, Veh, or POE astrocyte inserts (*n* = 90 total neurons per condition from three independent experimental replicates; * *p* < 0.05, **** *p* < 0.0001, Kruskal–Wallis test with Dunn’s multiple comparisons). (**C**) Quantification of neuronal branch complexity (*n* = 60 total neurons per condition from two independent experimental replicates; n.s. = not significant, ** *p* < 0.01, **** *p* < 0.0001, ANCOVA and pairwise comparisons with Bonferroni correction).

**Figure 3 cells-13-00837-f003:**
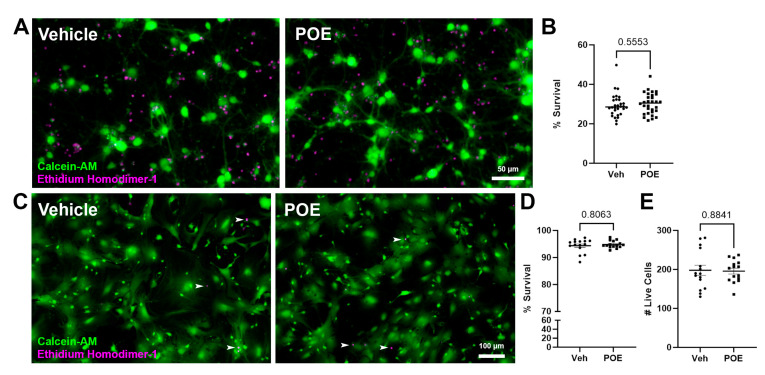
No effects of POE on cell survival and proliferation. (**A**) Vehicle control and POE mouse-derived cortical neurons after 12 days in astrocyte-free culture. Live neurons were stained with calcein-AM (green), while dead neurons were visualized via ethidium homodimer-1 (magenta). (**B**) Percentage of surviving neurons (= # live cells/# total cells) for vehicle or POE treatment (*n* = 30 total images per condition from two independent experimental replicates; unpaired *t*-test). (**C**) Vehicle control and POE mouse-derived cortical astrocytes after 10 days in culture. Live/dead staining is as in (**A**) (rare examples of dead astrocytes indicated with white arrowheads). (**D**,**E**) Percentage of surviving astrocytes (**D**) and number of live astrocytes (**E**) for vehicle or POE treatment (*n* = 15 total images per condition from two independent experimental replicates; Mann–Whitney test (**D**) or unpaired *t*-test (**E**)).

**Figure 4 cells-13-00837-f004:**
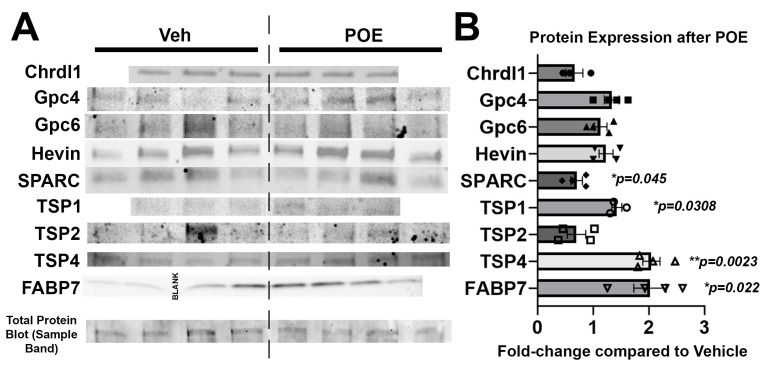
Changes in synapse-related protein expression in POE astrocytes. (**A**) Western blot images reveal expression patterns of prominent synapse-associated proteins from the DIV10 vehicle and POE astrocyte lysates. Bottom row shows a representative band from one of the total protein blot images that was used for normalization purposes (full image shown in Appendix A). (**B**) Quantification of Western blots for proteins indicated in (**A**). Data shown as fold-change in band intensity values for POE astrocyte lysates compared to vehicle (*n* = 3–4 experimental replicates; unpaired *t*-test (unlabeled *p* values were not significant)).

**Figure 5 cells-13-00837-f005:**
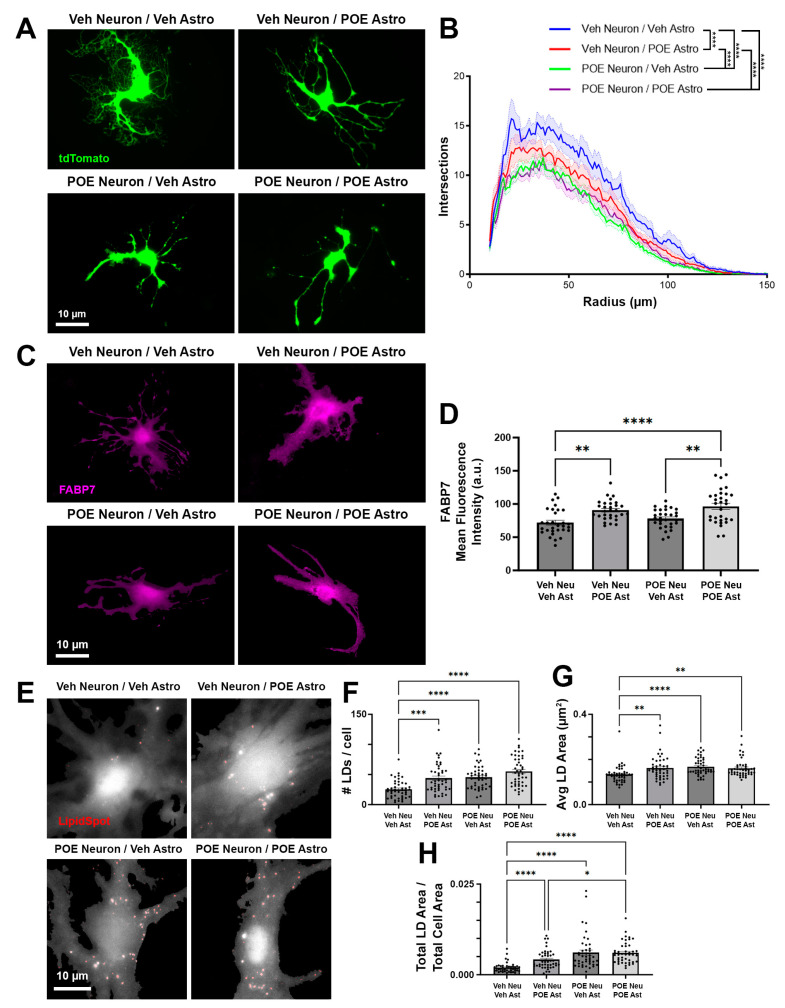
Disrupted astrocyte morphology and lipid content after POE. (**A**) Astrocytes isolated from Veh or POE mice. tdTomato-expressing astrocytes (green) were grown on top of Veh or POE neurons (not visible) for 48 h and fixed at DIV12. (**B**) Quantification of astrocytic branch complexity (*n* = 94–103 total astrocytes per condition from two independent experimental replicates; **** *p* < 0.0001, ANCOVA and pairwise comparisons with Bonferroni correction (unlabeled comparisons were not significant)). (**C**) FABP7 expression (magenta) in Veh or POE astrocytes cultured together with Veh or POE neurons. (**D**) Mean fluorescence intensity of FABP7 signal within the imaged astrocytes (*n* = 30 total astrocytes per condition from two independent experimental replicates; ** *p* < 0.01, **** *p* < 0.0001, one-way ANOVA with Tukey’s multiple comparisons). (**E**) Grayscale images of Veh and POE LipidSpot-stained astrocytes overlaid with the isolated LDs (red). (**F**–**H**) Quantification yielded the total number of LDs per cell (**F**), average area of individual LDs (**G**), and a ratio of the combined area of all LDs within a cell divided by the total area occupied by that cell (**H**) (*n* = 45 total astrocytes per condition from two independent experimental replicates; * *p* < 0.05, ** *p* < 0.01, *** *p* < 0.001, **** *p* < 0.0001, Kruskal–Wallis test with Dunn’s multiple comparisons).

## Data Availability

The raw data supporting the conclusions of this article will be made available by the authors on request.

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
