# Peer review of "Abnormal Morphology and Synaptogenic Signaling in Astrocytes Following Prenatal Opioid Exposure"

_cells, 2024, doi:10.3390/cells13100837_

Round 1
Reviewer 1 Report
Comments and Suggestions for Authors
The proposed manuscript titled " Abnormal Morphology and Synaptogenic Signaling in Astrocytes Following Prenatal Opioid Exposure " aims to investigate how the partial μ-opioid receptor agonist, buprenorphine, impacts astrocyte synaptogenic signaling and morphological development in cortical cell cultures.
Through a meticulously conducted study, the authors provided a comprehensive analysis of the synaptogenic signaling and morphological development of cortical cell population after prenatal opioid exposure. The study assessed the impact of prenatal opioid exposure on cellular and synaptic development during critical early stages of neural network formation. The findings revealed that the opioid buprenorphine disrupts the formation of excitatory synapses mediated by astrocytes between cortical neurons. A study of particular importance in addressing the widespread issue of drug use today, particularly during gestational phases.
Acknowledging the translational implications of their discoveries for clinical trials, the authors emphasized the importance of comprehending these molecular mechanisms across diverse diseases and therapeutic approaches, spanning from regenerative medicine to cancer therapy.
The materials and methods are meticulously detailed, and the results are reported and presented impeccably.
The conclusions drawn succinctly encapsulated the empirical evidence garnered from their study.
Additionally, the references cited were aptly selected, reflecting contemporary literature pertinent to the research domain.
In my opinion, there are no points that the authors should improve, and the work could be accepted in its current form.
Author Response
We thank the Reviewer for their encouraging review and recommendation of our manuscript.
Reviewer 2 Report
Comments and Suggestions for Authors
This study extends important findings from a previous work of the group (Boggess et al 2021) further uncovering the effects of prenatal opioid exposure on astrocytes and their interactions with neurons during development. The authors used an in vitro culture system to assess effects of buprenorphine on neurons cultured alone or together with astrocytes. An in vivo prenatal treatment paradigm (POE) was also used to compare the interactions between neurons and astrocytes isolated after prenatal administration of buprenorphine in pregnant dam. Upon opioid exposure, the authors observed significant changes in excitatory synapse number, which were accompanied by morphological changes in both neurons and astrocytes, as well as expression levels of astroglial proteins important for synaptogenesis and astroglial stress response. Intriguingly, the authors have also observed a complex neuroadaptive response where astrocytes from POE mice somehow retained some synaptogenic effects on neurons from POE mice but not on those from control mice. Overall, the study is interesting, well-designed and explained. It opens up future research directions towards mechanistic insights underlying the role of astrocytes in drug abuse and addiction. In particular, the in vitro culture system could be further exploited as a means to test for astroglial secreted factors as well as screen for potential drug targets. Please see following questions and suggestions:
1) The authors found no change in cell survival and proliferation in neurons and astrocytes between vehicle control and POE mice after 12 and 10 days in culture, respectively. Is it possible that cell death could occur acutely in vivo before isolation of cells and that these potentially dying or unhealthy cells did not survive the isolation and preparation process and thus not detected in this assay? Please comment.
2) Early administration of buprenorphine at E7 may also have direct impact on the proliferation and properties of progenitor cells that eventually give rise to both cortical neurons and astrocytes. Could the authors please discuss whether this is a possibility and postulate whether this could be contribute to the observed neuroadaptive response between neurons and astrocytes in POE mice?
3) Following up on the previous question, has there been work done to compare the transcriptomic states of neurons and astrocytes immediately after isolation from POE mice at P0-P1? Please discuss.
4) The authors detected an increase in synapse number upon astrocyte co-culture from control mice but not (at least to the same extent) those from POE mice (Figure 2B). The authors, however, also found corresponding decrease in neuronal morphological complexity (Figure 2C). Can the authors please explain how they can conclude that the decrease in synapse number directly reflects a decrease in synaptogenesis but not indirectly due to a decrease in neuronal complexity and size? Is the density of synapse different? Please clarify.
5) The authors used western blotting to detect the expression of several secreted proteins in astrocyte lysates obtained from control vs POE mice. Some of the proteins assessed show very faint and non-specific signals (e.g. Gpc4, Gpc6, and Tsp2). It is difficult to interpret whether this reflects a genuine lack of significant difference between experimental groups. Please consider replacing with different blots, repeating with different antibodies or removing these proteins from the analysis.
6) Following up on the previous question, have the authors tried or can they comment on whether assessing the culture medium for the presence of these secreted proteins would be appropriate?
Reviewer 3 Report
Comments and Suggestions for Authors
Niebergall, Weekley and coworkers examine how a partial μ-opioid receptor agonist buprenorphine affects astrocyte synaptogenic signaling and morphological development in cortical cell culture. The experiments were performed by using immunocytochemistry, Western blotting, imaging analysis and so on. As a result, it was found that buprenorphine interferes with the process of astrocyte-mediated excitatory synapse formation between cultured cortical neurons. It was suggested that prenatal opioid exposure pathology may be caused by impaired glial-mediated synaptic development. Although this manuscript is well-written and seems to be interested, there are several points that could help to improve this manuscript, as follows:
Major points:
1. This manuscript uses many abbreviations, making it a little difficult to read. Therefore, an alphabetically ordered list of abbreviations should be provided.
2. Whenever possible, abbreviations should be defined on their first occurrence. For instance, thrombospondine is defined in line 56, not in line 408. BDNF is defined in line 119, not in line 521. PBS is defined in line 101, not in line 175.
3. The same academic terminology should be used throughout the text and figures. For example, “hevin” in line 56 while “Hevin” in line 215; “Tsp” in lines 216 and 217 while “TSP” in line 506; “FABP7” in line 411 while “Fabp7” in Fig. 4. This point should be amended.
4. Do the results obtained by using buprenorphine in this manuscript also apply to the other opioids? Also, are the results obtained mediated by activation of opioid receptors? Discuss these points if possible.
5. References: each letter in the title of the reference begins (Ref. 24) or does not begin (Ref. 25) with a capital letter. All of the references given should be consistent in style. Please amend this point.
Specific points:
1. Line 103: “ovomucoid inhibitor” seems to be a little vague, because “ovomucoid” is a trypsin inhibitor. Please amend this point.
2. Line 118: please define “Pen/Strep”.
3. Line 119: please expand “CNTF”.
4. Lines 123, 144, 158, 165, 166, 179 and so on: please subscript the numbers.
5. Lines 128 and 129: although buprenorphine (partial μ-opioid receptor agonist; see line 17) is used at 500 nM, is it possible that buprenorphine activates not only μ- but also δ- and κ-opioid receptors? Please make this point clear.
6. Lines 157 and 185: please use either “hr” (line 157) or “hours” (line 185) throughout the manuscript.
7. Lines 190 and 195: there seems to be no statement about the results obtained using “DAPI” in 3. Results. Please amend this point.
8. Line 212: please state clearly how “Chrdl1” is related to “chordin-like 1” in line 57.
9. Lines 225, 226 and 439: please put a space between value and unit. Please use either “minutes” or “min” throughout the manuscript.
10. Lines 241 and 242: what are the unites of “50” and “4”? Please amend this point.
11. Line 258: please define “LD” here, not in line 467, because “lipid droplet” appears first in line 258 in the text.
12. Lines 296 and 297: gabapentin is known to inhibit voltage-gated Ca2+ channels. Is the effect mentioned here due to voltage-gated Ca2+ channel inhibition? Please make this point clear.
13. Line 392: there is no statement about “calcein-AM” and “ethidium homodimer-1” in Section 2.6. Please amend this point.
14. Line 467: the definition of “LD” should be consistent throughout the manuscript. Please see this line and line 26.
15. Line 519: “(ACM)” may be unnecessary, because this seems to only appear on this line.
16. Line 529: what about glycinergic synapses? Please see DOI: 10.1113/jphysiol.1991.sp018696.
17. Line 534: not “suggests” but “suggest”? Please check English.
18. Line 566: it is not necessary to define “LD” here. Please see line 467.
19. There may be more mistakes than pointed above. Please check your manuscript very carefully from scientific points of view.
Comments on the Quality of English LanguageEnglish language is fine.
